# Does Income Inequality Create Excessive Threats to the Sustainable Development of Russia? Evidence from Intercountry Comparisons via Analysis of Inequality Heatmaps

**Mikhail Lvovitch Dorofeev** 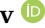

Public Finance Department, Financial Faculty, Financial University under the Government of the Russian Federation, 49 Leningradsky Prospekt, 125993 Moscow, Russia; dorofeevml@yandex.ru

**Abstract:** The paper explored the problem of income inequality in Russia in the context of the sustainable development of Russia. The research starts from the historical analysis of income inequality dynamics in Russia. Then, we discussed the problem of the inconsistency of data, comparing different sources (official data from the Rosstat database and alternative data from the World inequality database). The purpose of this research was to assess Russian specifics of income inequality and answer the question of if the income inequality in Russia is excessively high and needs extra government regulation in order to reach the trajectory of advanced sustainable development. To this end, we made intercountry comparisons and used the method of building income inequality heatmaps basing on a dataset from the World Inequality Database. Our sample includes the per-adult equivalent of household market income distribution in 27 developed and developing countries and world regions. The result of the research was that there are many countries in the world wherein the differentiation of income exceeds Russia's. Russian income inequality is lower than the world average, but the structure of the Russian household income distribution stands out by an extreme concentration of national income in the hands of the top 1%. We supported our results via the independent data from the Credit Suisse wealth inequality report, connecting a record level of wealth inequality in Russia with its problem of top 1% income inequality. It is recommended to gradually increase marginal tax rates on the income and wealth of the top 1% and continue developing an effective progressive tax system in Russia.

**Keywords:** economic inequality; poverty; socio-economic policies; government regulation; taxes; fiscal policy; sustainable development

## 1. Introduction

Socio-economic inequality refers to differences between individuals in a group, between population groups, or between countries according to such criteria as: (1) household expenditures on maintaining a specific standard of living, (2) the level of income (usually labor income and income from the use of property, including financial assets), and (3) the level of wealth (savings owned and managed by the household).

Long-term changes in inequality are influenced by a combination of political, economic, and social shocks. These shocks typically lead to changes in public sentiment, expectations, and behavior of investors, as well as the corresponding reforms of financial regulators. Financial crises make regulators change the configuration of the financial mechanism of the economy. In turn, these reforms usually influence the redistribution of national income between households and lead to a change in the long-term trend in the dynamics of economic inequality. By adjusting the financial mechanism of the economy and the redistribution of national income, we mean the parameters of fiscal and monetary policy that can influence the pace of change in income and wealth of various groups of the population (Dorofeev 2020).

The progress in trade globalization and technological development of a country is associated with reduction in income inequality, while financial globalization and foreign direct investment inflows are, on the contrary, associated with an increase in inequality (Sánchez López et al. 2019).

Social justice is the most obvious goal of combating economic inequality. The creation of equal opportunities and an equal chance of prosperity for everyone is the basic requirement for building a social state. Sociologists believe that economic inequality can be acceptable even if there are super-rich people in society, provided that poverty is completely eliminated (Peterson Institute for International Economics 2020). At the same time, economists primarily consider the issue of regulating socio-economic inequality in the context of the search for a compromise between inequality and economic growth (Shevyakov 2011; Kostyleva 2011; Varsavsky 2016, 2019; Rossoshansky 2019; Dorofeev 2020; Morozko et al. 2020, 2021).

There is significant empirical evidence suggesting that high levels of socio-economic inequality can be harmful for economic growth and sustainable development (Alesina and Rodrik 1994; Persson and Tabellini 1994; Voitchovsky 2005; Castelló-Climent 2010; Cingano 2014; Halter et al. 2014). The higher the level of economic inequality, the more problems this can create for the growth of the economy. Socio-economic inequality can negatively affect economic growth through imperfections of the credit market, disruption of socio-political stability, restriction of investment in human capital, an increase in risks for investors, and the blocking of valuable development initiatives and reforms, etc. (Forbes 2000; Knowles 2005; Tridico and Meloni 2018; Dorofeev 2020). In most EU countries, only a minority of people benefited from economic growth, resulting in increased inequality and poverty (Michálek and Výbošťok 2018). From this point of view, inequality can indeed become a destabilizing factor which slows down economic growth and may even lead to a recession.

Most emerging economies show levels of income inequality higher than in the five most unequal OECD countries, while the picture the picture is more mixed when it comes to inequalities in other dimensions of people's well-being (Balestra et al. 2018).

Is it possible to argue that today we know a lot about economic inequality in Russia? The answer is not as obvious and simple as it may seem at first glance. The study of socio-economic inequality in modern Russia is a very relevant issue for a number of reasons. Firstly, this is due to the collapse of the USSR and the ambiguous influence of this event on the formation of the Russian market economy in the context of opportunity inequalities during privatization reforms in the early 1990s. Secondly, because of great discrepancies in estimates of the historical and current levels of economic inequality, represented in different sources. Thirdly, in the context of the increase of social tension associated with noticeable wealth differentiation of Russian households.

A review of literature concerning the analysis of economic inequality in Russia showed that there are alternative estimates of income inequality of Russian households. They differ significantly from official Rosstat reports and data (Rosstat 2021a, 2021b). According to them, the level of inequality in Russia is almost two times higher than reported in the official data (Kostyleva 2011; Varsavsky 2016, 2019; Matytsin and Ershov 2012; Ovcharova et al. 2016; Shevyakov 2010, 2011; Livshits and Livshits 2017, 2018). This point of view is rather debatable, and has many critics.

Kapeliushnikov (2020) heavily criticized the quality of statistics from the World inequality database used for historical analysis of economic inequality in Russia in a widely known article by Novokmet et al. (2018). Kapeliushnikov (2020) mentioned methodological artifacts, inconsistencies, and controversial questions left without answers in the research of T. Piketty's team. He concluded that no assessment of the economic inequality can be considered as an "objective fact", since this assessment is overwhelmingly preceded by a large number of conventions, calculations, assumptions, extrapolations, and other methodological manipulations. Changing the method of obtaining this "objective fact" increases the probability of obtaining "other valuations" of economic inequality. Does this mean that it is better to abandon the idea of searching for the correct quantitative assessment of income inequality in Russia? From our

point of view, the answer is "definitely not", because this work is very important for building a social state in Russia as an integral part of a prosperous world economy.

Balatsky and Ekimova (2018) also criticized the point of view that Russia has an extremely high level of inequality calculated on the basis of the World inequality database. They pointed out that the decile coefficient of funds calculated with the data from World inequality database is 13 times higher than the estimates of Rosstat and almost 19 times higher than the estimates of the World Bank. It is regarded as an over-interpretation of the problem of income inequality in Russia and as being prone to extremes. However, we cannot accept their criticism and arguments, because Rosstat data are formed with a different methodology and are not consistent with any other international databases, such as the World Inequality Database (2020b). That is the reason why (1) the decile coefficient of funds calculated with different data is also different; (2) to acquire the correct international valuations, we used the unified and consistent data for our calculations.

Income inequality plays a special role among others types of inequality because it has a decisive impact on the current standard of living and future level of household wealth. That is the first reason why we studied income inequality in this research to understand the level of economic inequality in Russia. Another important reason is the poor quality and unavailability of wealth inequality data on majority of countries, meaning we cannot make intercountry comparisons based on this data.

The purpose of this study was to assess the level of economic inequality in Russia based on various information sources with household income distribution and conduct intercountry comparisons of Russia with other countries. This will help to understand whether inequality threats the sustainable development of Russia and provide recommendations for state financial regulation of this problem.

## 2. The History of Income Inequality in Russia

The most dynamic period in the changes in the level of income inequality in Russia was the last decade of the last century, starting with the collapse of the USSR. In the early and middle USSR, real incomes of the lower 90% grew faster than real incomes of the upper 10% of households, and this can be considered as a great achievement of the USSR's government economic policy (Table 1).

**Table 1.** The average annual growth rate of real pre-tax household incomes in Russia during the period of 1905–2016.

| Group/Percentile | 1905–1956 | 1956–1989 | 1989–2016 | 1905–2016 |
|---|---|---|---|---|
| The entire population | 1.90% | 2.50% | 1.30% | 1.90% |
| Lower 50%, incl. | 2.60% | 3.20% | −0.80% | 1.90% |
| 10 p | 2.76% | 2.86% | −2.58% | 1.52% |
| 20 p | 2.58% | 3.12% | −0.70% | 1.96% |
| 30 p | 2.56% | 3.17% | −0.43% | 2.03% |
| 40 p | 2.62% | 3.05% | −0.25% | 2.07% |
| 50 p | 2.77% | 2.81% | −0.12% | 2.10% |
| Middle 40%, incl. | 2.50% | 2.30% | 0.50% | 2.00% |
| 60 p | 2.71% | 2.50% | 0.12% | 2.05% |
| 70 p | 2.64% | 2.23% | 0.36% | 1.99% |
| 80 p | 2.47% | 2.18% | 0.77% | 1.99% |
| 90 p | 1.73% | 2.09% | 1.37% | 1.75% |
| Top 10%, incl. | 0.80% | 2.30% | 3.80% | 1.90% |
| 91 p | 1.67% | 2.10% | 1.66% | 1.79% |
| 95 p | 1.29% | 2.21% | 2.58% | 1.85% |
| 99 p | 0.50% | 2.40% | 3.34% | 1.70% |
| Top 1%, incl. | −0.30% | 2.50% | 6.40% | 2.00% |
| 99.1 p | 0.46% | 2.40% | 3.45% | 1.70% |
| 99.5 p | 0.19% | 2.42% | 4.42% | 1.80% |
| 99.9 p | −0.44% | 2.52% | 6.17% | 1.91% |
| Top 0.1%, incl. | −1.20% | 2.70% | 9.50% | 2.30% |
| 99.99 p | −2.10% | 3.00% | 12.20% | 2.50% |
| 99.999 p | −3.00% | 3.30% | 14.90% | 2.70% |

Source: compiled by the author based on the World Inequality Report (2018).

Despite the low level of socio-economic inequality in the USSR comparedwith modern Russia, it was still relatively higher than in other communist countries (Czech Republic, Poland, Hungary, China, etc.). In addition, after the collapse of the USSR, it skyrocketed and became twice as high as in other post-communist countries (Figure 1).

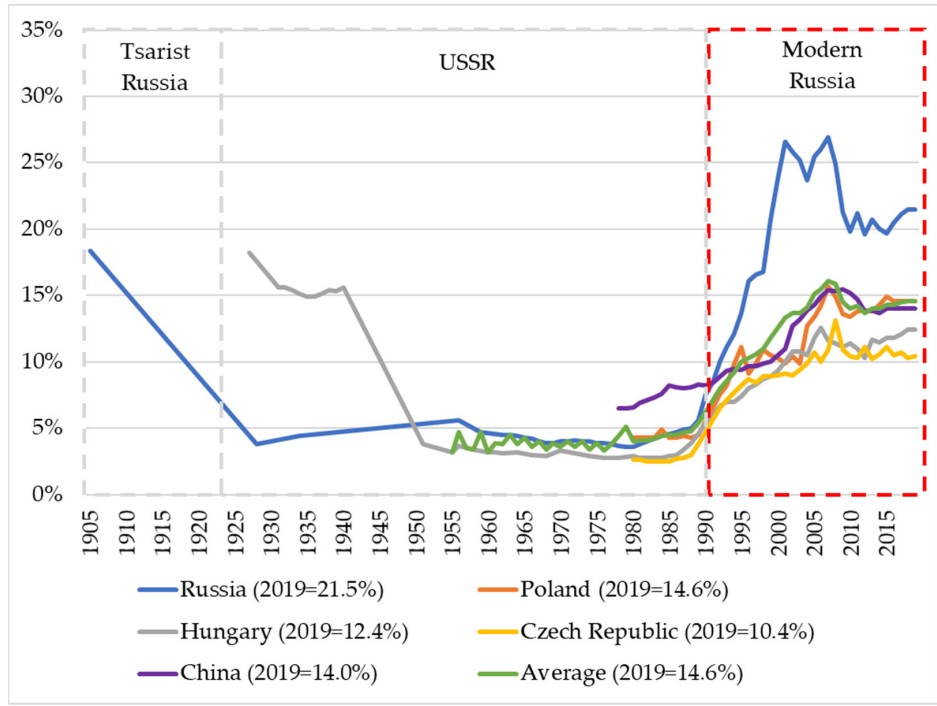

**Figure 1.** Pre-tax income shares of the top 1% of households in Russia and other countries where an experiment of building communism was conducted. Period of 1905–2019. Source: Compiled by the author based on the World Inequality Database (2020a).

The growth rate of real incomes of Russian households from the upper and lower parts of the distribution in the period 1989–2016 diverged dramatically. Real income and the fortunes of rich households in Russia were increasing much more rapidly than of other population between the collapse of USSR and The Great Recession. The transitionary period and shaping market economy in Russia, unfair privatization campaigns, the emergence of financial market institutions, and other factors created the period of unequal opportunities in Russia when income inequality reached its maximum levels in the last 100 years.

After the collapse of the USSR, the national per capita income for PPP in Russia dramatically fell and did not recover to the levels of the 1970s until now (Table 2).

**Table 2.** National per capita income for PPP in Europe, 2019, calculated as % of the global average.

| Year | Russia | China | USA | France | German | Japan | BRICS | Europe | World |
|------|--------|-------|-----|--------|--------|-------|-------|--------|-------|
| 1970 | 157.0% | 19.6% | 292.9% | 239.0% | 270.3% | 156.2% | 89.2% | 192.2% | 100.0% |
| 1980 | 163.3% | 19.6% | 274.9% | 254.4% | 281.3% | 181.9% | 96.5% | 202.6% | 100.0% |
| 1990 | 162.4% | 25.7% | 314.8% | 276.5% | 304.7% | 238.1% | 88.1% | 217.7% | 100.0% |
| 1995 | 87.4% | 35.3% | 350.0% | 286.4% | 284.0% | 239.8% | 75.6% | 224.9% | 100.0% |
| 2000 | 106.0% | 36.3% | 380.8% | 296.4% | 279.6% | 214.5% | 75.3% | 233.4% | 100.0% |
| 2005 | 130.3% | 48.2% | 370.3% | 278.2% | 262.5% | 203.2% | 82.1% | 227.7% | 100.0% |
| 2010 | 138.1% | 64.5% | 331.9% | 253.7% | 251.5% | 184.7% | 88.6% | 212.8% | 100.0% |
| 2015 | 134.9% | 81.0% | 327.2% | 229.3% | 241.4% | 184.3% | 87.5% | 197.5% | 100.0% |
| 2016 | 127.6% | 84.4% | 320.7% | 226.7% | 242.5% | 182.1% | 85.4% | 197.8% | 100.0% |
| 2017 | 126.7% | 87.1% | 318.4% | 226.0% | 242.0% | 182.5% | 85.1% | 198.6% | 100.0% |
| 2018 | 128.4% | 90.1% | 315.2% | 224.0% | 240.8% | 181.3% | 85.3% | 199.2% | 100.0% |
| 2019 | 128.4% | 92.5% | 316.9% | 224.4% | 241.7% | 182.0% | 85.4% | 200.7% | 100.0% |

Source: compiled basing on World Inequality Database (2020a).

The USSR experiment with the construction of communism and the prohibition of private property was indeed unprecedented. Its natural consequence was a significant decrease in the level of household income inequality, as we can see even from official statistics of Rosstat.

There is significant evidence that the leaders of the Communist Party had special privileges and opportunities that the ordinary citizen of the USSR did not have (Zubkova 2013). This allowed them to increase their actual standard of living 4–5 times higher than what they could afford on their official income. So, based on the definition of social-economic inequality, we can say that economic inequality in the USSR was higher than the official valuations. The huge inequality of opportunities in the USSR contrasted strongly with the official "equality" of income. Ordinary Soviet citizens who received lower disposable incomes than communist party leaders often could not even spend their money. This was hindered by the high-quality consumer goods deficit in the country. In other words, on paper, the level of economic inequality was low, but in fact this did not mean that the USSR had a high standard of living for households and equal opportunities for all its citizens.

The political regime in Russia has changed several times in the period between 1905–2016. The system of statistical and tax accounting during this period went through many difficult stages of development. That is why we should keep in mind the risks of dealing with a low quality of statistical data when analyzing the economic history of USSR. Such risks are typical for any undemocratic country with (1) a dictatorial type of government, (2) the lack of an institution of free media, and (3) a low level of government accountability, etc.

The rapid increase of income inequality in the period of 1992–2008, which reaching 100-year maximums in 2008, slowed down the economic growth of Russia significantly after The Great Recession. In order to boost economic growth in Russia, it is necessary to understand how high the level of income inequality is and whether the government should take urgent measures to reduce it. This can be achieved through an intercountry analysis of the income inequality of Russia as compared with other countries.

## 3. Methodology

The research was conducted on the basis of Rosstat database (Rosstat 2021a) and the World Inequality Database (2020a). The first of them was necessary to clarify and assess the completeness and transparency of Rosstat data and specify an idea of an official point of view on the problem of economic inequality in Russia. The World inequality database was selected as a basis for intercountry comparisons of income inequality in Russia. We understand the risk of under- or over- estimation the level of income inequality in Russia if we use information not from the official Rosstat database. However, the problem is that Rosstat data are not suitable for conducting intercountry comparisons, since they are badly comparable to any other international database, such as LIS (2021), OECD (2021), The World Bank (2021), etc.

Common disadvantages of any database for the purposes of our research except the World inequality database, are the following: (1) the inaccessibility of primary data to perform the required calculations on our analysis methodology; (2) lots of passes; (3) poor coverage of historical retrospectives; (4) different methodology of accounting and classification of household income distribution, etc. In this regard, we had to use the data from World Inequality to make correct international comparisons of income inequality in Russia.

To conduct the study, we formed a sample of 27 countries on the basis of the World Inequality Database, including BRICS, the largest countries in Europe, the largest English-speaking countries, as well as three world regions (Asia, Europe, and the European Union) and the world economy as a whole. This 27-county sample excluded a lot of countries with low income per capita and, in general, represents countries with high and average income per capita. This specific choice of counties helped us to compare Russia with more similar countries in terms of size of the economy or the structure of its exports.

Our sample consisted of pre-tax data of national income distribution. Income was calculated as an equivalent (equal split size) per adult of household. The list of the symbols and coefficients used in the data analysis is shown in Table 3.

**Table 3.** System of indicators and ratios for analysis of income inequality in Russia.

| № | Share/Ratio | Description of Household Group/Ratio |
|---|---|---|
| 1. | p0p10 [1] | Lower 10% |
| 2. | p0p20 | Lower 20% |
| 3. | p0p40 | Lower 40% |
| 4. | p0p50 | Lower 50% |
| 5. | p50p90 | Average 40% |
| 6. | p80p100 | Top 20% |
| 7. | p90p100 | Top 10% |
| 8. | p99p100 | Top 1% |
| 9. | p90p100/p0p10 | Decile ratio (10% of the richest to 10% of the poorest). |
| 10. | p90p100/p0p40 | Palm ratio (ratio of 10% of the richest to 40% of the poorest). |
| 11. | p90p100/p0p50 | Income ratio between the richest 10% and the poorest 50%. |
| 12. | p50p90/p0p50 | Income ratio between 40% of the middle class and 50% of the low-income households. |
| 13. | p80p100/p0p20 | Quintile ratio (20% of the richest to 20% of the poorest). |
| 14. | p80p100/p0p50 | The income ratio between the richest 20% and the poorest 50%. |
| 15. | p99p100/p90p100 | Share of 1% of super-rich people in the upper decile group. |

[1] p in p0p10 income group is a percentile and a range boundary of this group.

The method of building heat maps is widely used in econophysics research based on the use of the wavelet analysis of the relationships between different macroeconomic indicators and economic inequality (Yakovenko 2009; Chang et al. 2018, 2019). In most cases, this method is used as a graphical tool to show the thermal spectrum of household income distribution or to demonstrate correlations of different indicators.

The inequality heatmap is a simple and informative tool that provides very detailed information and a deeper understanding about household income distribution than any particular econometric model. At the same time, it has some limitations connected with its bulkiness and lengthiness compared with slender and concise econometric models.

The official point of view of Rosstat on income inequality in Russia is given in the format of a heatmap and additional calculations of statistical averages. To conduct inter-country comparisons of income inequality in Russia, we used a comparative method of assessment, building heatmaps of household income distributions with the use of data from the last 10 available years, from 2009 to 2019. In the field of inequality heatmaps, we showed the data for the beginning and the end of this period as well as 10-year averages. Heatmaps were built in MS Excel. Red (hot) color indicates maximum and blue (cold) color indicates minimum levels of inequality.

## 4. Results

### 4.1. Income Inequality of Russian Households: Official Point of View

Rosstat methodology shows that after the collapse of the USSR, the share of pre-tax incomes of the top 20% of households increased significantly and fixed at the level of 46%−48% (Table 4). This is almost half of all income earned in the country, which is noticeably different from the period of 1970–1990, when the same group of people received about a third of all national income. This section may be divided by subheadings. It should provide a concise and precise description of the experimental results, their interpretation, as well as the experimental conclusions that can be drawn.

**Table 4.** Income inequality in Russia, Rosstat database.

| Year | Distribution of Household Income by 20% Income (%) | | | | | Decile Ratio | Gini Index |
|---|---|---|---|---|---|---|---|
| | 0–20 | 20–40 | 40–60 | 60–80 | 80–100 | | |
| 1970 | 7.8 | 14.8 | 18.0 | 22.6 | 36.8 | n/a | n/a |
| 1980 | 10.1 | 14.8 | 18.6 | 23.1 | 33.4 | n/a | n/a |
| 1990 | 9.8 | 14.9 | 18.8 | 23.8 | 32.7 | n/a | n/a |
| 1995 | 6.1 | 10.8 | 15.2 | 21.6 | 46.3 | 13.5 | 0.387 |
| 1996 | 6.1 | 10.7 | 15.2 | 21.6 | 46.4 | 13.3 | 0.387 |
| 1997 | 5.9 | 10.5 | 15.3 | 22.2 | 46.1 | 13.6 | 0.390 |
| 1998 | 6.0 | 10.6 | 15.0 | 21.5 | 46.9 | 13.8 | 0.394 |
| 1999 | 6.0 | 10.5 | 14.8 | 21.1 | 47.6 | 14.1 | 0.400 |
| 2000 | 5.9 | 10.4 | 15.1 | 21.9 | 46.7 | 13.9 | 0.395 |
| 2001 | 5.7 | 10.4 | 15.4 | 22.8 | 45.7 | 13.9 | 0.397 |
| 2002 | 5.7 | 10.4 | 15.4 | 22.7 | 45.8 | 14.0 | 0.397 |
| 2003 | 5.5 | 10.3 | 15.3 | 22.7 | 46.2 | 14.5 | 0.403 |
| 2004 | 5.4 | 10.1 | 15.1 | 22.7 | 46.7 | 15.2 | 0.409 |
| 2005 | 5.4 | 10.1 | 15.1 | 22.7 | 46.7 | 15.2 | 0.409 |
| 2006 | 5.3 | 9.9 | 15.0 | 22.6 | 47.2 | 15.9 | 0.415 |
| 2007 | 5.1 | 9.8 | 14.8 | 22.5 | 47.8 | 16.7 | 0.422 |
| 2008 | 5.1 | 9.8 | 14.8 | 22.5 | 47.8 | 16.6 | 0.421 |
| 2009 | 5.2 | 9.8 | 14.8 | 22.5 | 47.7 | 16.6 | 0.421 |
| 2010 | 5.2 | 9.8 | 14.8 | 22.5 | 47.7 | 16.6 | 0.421 |
| 2011 | 5.2 | 9.9 | 14.9 | 22.6 | 47.4 | 16.2 | 0.417 |
| 2012 | 5.2 | 9.8 | 14.9 | 22.5 | 47.6 | 16.4 | 0.420 |
| 2013 | 5.2 | 9.9 | 14.9 | 22.6 | 47.4 | 16.1 | 0.417 |
| 2014 | 5.3 | 9.9 | 15.0 | 22.6 | 47.2 | 15.8 | 0.415 |
| 2015 | 5.3 | 10.1 | 15.0 | 22.6 | 47.0 | 15.5 | 0.412 |
| 2016 | 5.3 | 10.1 | 15.0 | 22.6 | 47.0 | 15.5 | 0.412 |
| 2017 | 5.3 | 10.1 | 15.1 | 22.6 | 46.9 | 15.4 | 0.411 |
| 2018 | 5.3 | 10.0 | 15.0 | 22.6 | 47.1 | 15.6 | 0.413 |
| 2019 | 5.3 | 10.1 | 15.1 | 22.6 | 46.9 | 15.4 | 0.411 |
| 2020 | 5.5 | 10.3 | 15.3 | 22.7 | 46.2 | 14.5 | 0.403 |
| Maximum | 10.1 | 14.9 | 18.8 | 23.8 | 47.8 | 16.7 | 0.422 |
| Average | 5.9 | 10.6 | 15.4 | 22.5 | 45.6 | 15.1 | 0.408 |
| Median | 5.4 | 10.1 | 15.1 | 22.6 | 46.9 | 15.4 | 0.411 |
| Minimum | 5.1 | 9.8 | 14.8 | 21.1 | 32.7 | 13.3 | 0.387 |

Source: Rosstat (2021a).

The prior role in characterizing income inequality according to Rosstat is devoted to market income per capita group of indicators. The level and dynamics of the decile ratio and Gini index, according to the Rosstat (2021a), indicate a rapid increase in income inequality in Russia from the moment of the collapse of the USSR until the Great Recession in 2008 (Figure 2). This trend has a high correlation with the dynamics of commodity markets prices. It is quite obvious that dynamics of income inequality are closely correlated with unequal opportunities of different household groups in sharing economic benefits arising from Russian hydrocarbon exports in the period of high oil prices.

The trend of rising income inequality in Russia decreased after a reversal of oil prices. After 2008, the top 20% household national income share began to decline. On the other hand, national income shares of the other four quintile household groups are moving up and restoring their losses of the early 2000s. The most stable share of national income during this period belongs to the group of 60%–80% (third quintile group).

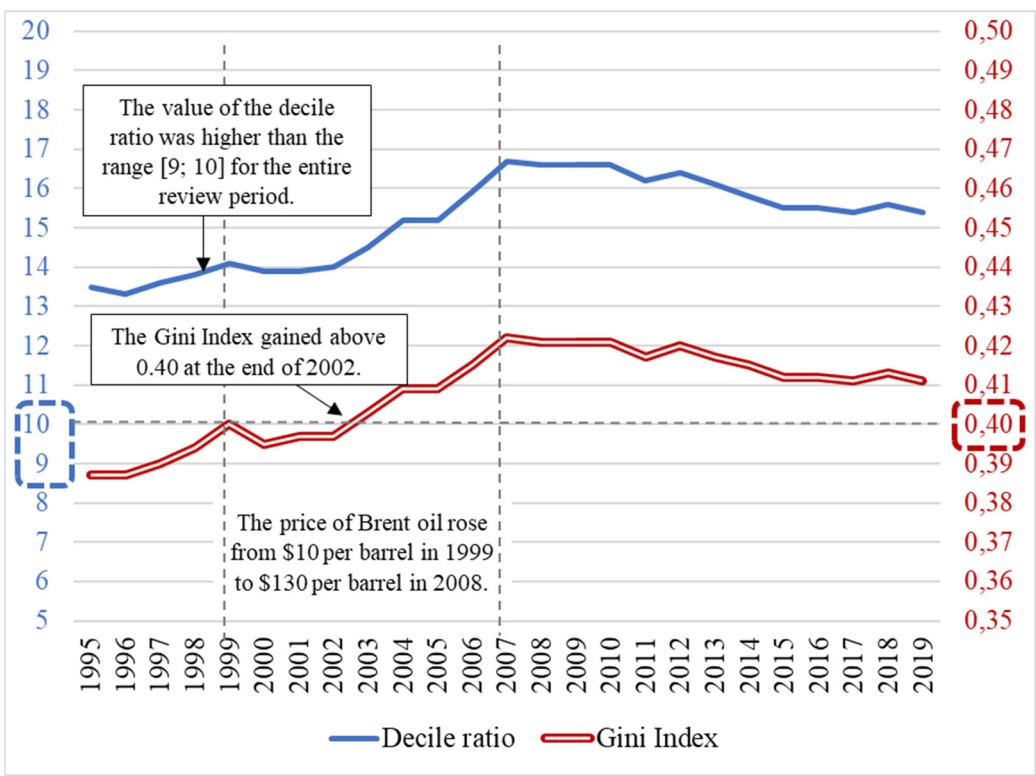

**Figure 2.** Dynamics of the decile ratio and the Gini coefficient. Source: Rosstat (2021a).

The maximum levels of income inequality in Russia were reached in the periods of economic crises in Russia (1998 and 2008). We can conclude that the welfare of poor households in Russia is very sensitive to economic crises. At the same time, the middle class (60–80 group) and rich Russians (80–100 income group) were living through a financial crisis rather successfully. The level of extreme wealth and the increase in the concentration of dollar billionaires in Russia during crises is only accelerating (Ovcharova et al. 2016; Credit Suisse Group AG 2020; Rossoshansky 2019).

There is a social query about moving towards to a more homogeneous wealth and income distribution in Russia. Various opinion polls and studies of public sentiment of Russian households show that despite the gradual decline of the decile ratio and the Gini index after 2008, Russian society still estimates the level of socio-economic inequality and poverty as high (Mareeva and Tikhonova 2016).

*4.2. Intercountry Comparisons of Income Inequality in Modern Russia: Alternative Point of View*

The structure of national income distribution of Russian households has much in common with countries such as Brazil, Israel, India, Qatar, Mexico, Turkmenistan, Turkey, and South Africa (Table 5). These countries have one common feature—they are part of the group of developing economies and most of them are commodity exporters. The main problem of economic inequality in all these countries is the extremely high concentration of income and wealth in the hands of the top 10% and especially the top 1% of households. At the same time, judging by the data of the World inequality database, Russia is not a country with the worst indicators of economic inequality.

**Table 5.** The distribution of pre-tax equivalent income by household income groups.

| № | Percentile | p0p10 (Bottom 10%) | | | p0p20 (Bottom 20%) | | | p0p40 (Bottom 40%) | | |
|---|---|---|---|---|---|---|---|---|---|---|
| | Country | 2009 | 2019 | 10Y Av. | 2009 | 2019 | 10Y Av. | 2009 | 2019 | 10Y Av. |
| | 1 | 2 | 3 | 4 | 5 | 6 | 7 | 8 | 9 | 10 |
| 1 | England | 0.29% | 0.35% | 0.32% | 2.23% | 2.74% | 2.53% | 10.97% | 12.93% | 12.18% |
| 2 | Brazil | 0.19% | 0.14% | 0.18% | 1.06% | 0.69% | 0.99% | 6.21% | 5.52% | 6.34% |
| 3 | Bulgaria | 0.33% | 0.28% | 0.30% | 2.57% | 2.22% | 2.38% | 12.40% | 10.36% | 11.41% |
| 4 | Germany | 0.31% | 0.30% | 0.30% | 2.44% | 2.35% | 2.34% | 12.01% | 11.61% | 11.63% |
| 5 | Denmark | 0.42% | 0.39% | 0.40% | 3.32% | 3.09% | 3.13% | 15.30% | 14.14% | 14.33% |
| 6 | Israel | 0.22% | 0.24% | 0.23% | 1.72% | 1.90% | 1.82% | 8.27% | 9.11% | 8.74% |
| 7 | India | 0.26% | 0.23% | 0.24% | 2.05% | 1.82% | 1.87% | 9.47% | 8.42% | 8.65% |
| 8 | Spain | 0.35% | 0.35% | 0.34% | 2.72% | 2.73% | 2.69% | 12.71% | 13.08% | 13.03% |
| 9 | Italy | 0.37% | 0.33% | 0.34% | 2.86% | 2.61% | 2.69% | 13.72% | 12.86% | 13.19% |
| 10 | Canada | 0.31% | 0.28% | 0.30% | 2.40% | 2.20% | 2.32% | 10.57% | 9.75% | 10.24% |
| 11 | Qatar | 0.21% | 0.23% | 0.23% | 1.63% | 1.78% | 1.75% | 7.78% | 8.11% | 8.04% |
| 12 | China | 0.20% | 0.20% | 0.20% | 1.58% | 1.60% | 1.58% | 8.36% | 8.52% | 8.43% |
| 13 | Luxembourg | 0.38% | 0.34% | 0.35% | 2.99% | 2.64% | 2.73% | 13.87% | 12.82% | 12.96% |
| 14 | Mexico | 0.18% | 0.23% | 0.21% | 0.82% | 1.03% | 0.91% | 4.17% | 5.00% | 4.49% |
| 15 | Netherlands | 0.40% | 0.36% | 0.38% | 3.11% | 2.84% | 2.96% | 14.91% | 13.82% | 14.23% |
| 16 | New Zealand | 0.35% | 0.33% | 0.33% | 2.72% | 2.60% | 2.63% | 12.50% | 11.98% | 12.08% |
| 17 | Norway | 0.45% | 0.43% | 0.43% | 3.51% | 3.36% | 3.38% | 16.19% | 15.45% | 15.61% |
| 18 | Poland | 0.33% | 0.33% | 0.33% | 2.60% | 2.59% | 2.58% | 12.51% | 12.54% | 12.51% |
| 19 | Romania | 0.23% | 0.23% | 0.18% | 1.82% | 1.77% | 1.67% | 9.31% | 9.13% | 9.58% |
| **20** | **Russia** | **0.24%** | **0.29%** | **0.28%** | **1.91%** | **2.29%** | **2.20%** | **9.04%** | **10.78%** | **10.45%** |
| 21 | USA | 0.19% | 0.19% | 0.18% | 1.50% | 1.46% | 1.44% | 7.99% | 7.77% | 7.63% |
| 22 | Turkmenistan | 0.20% | 0.20% | 0.20% | 1.60% | 1.60% | 1.58% | 8.01% | 8.01% | 8.03% |
| 23 | Turkey | 0.25% | 0.28% | 0.27% | 1.99% | 2.16% | 2.13% | 9.30% | 9.98% | 9.85% |
| 24 | Finland | 0.38% | 0.38% | 0.38% | 2.99% | 3.01% | 2.97% | 13.88% | 13.95% | 13.85% |
| 25 | France | 0.35% | 0.34% | 0.34% | 2.73% | 2.68% | 2.70% | 13.59% | 13.48% | 13.52% |
| 26 | Czech Republic | 0.49% | 0.47% | 0.48% | 3.85% | 3.66% | 3.77% | 16.95% | 16.39% | 16.62% |
| 27 | Sweden | 0.41% | 0.42% | 0.42% | 3.24% | 3.33% | 3.28% | 15.37% | 15.62% | 15.41% |
| 28 | South Africa | 0.10% | 0.05% | 0.06% | 1.23% | 0.67% | 0.78% | 5.68% | 3.42% | 3.87% |
| 29 | Japan | 0.32% | 0.31% | 0.31% | 2.48% | 2.44% | 2.45% | 11.44% | 11.28% | 11.30% |
| 30 | Asia | 0.15% | 0.15% | 0.15% | 1.39% | 1.41% | 1.38% | 6.27% | 6.35% | 6.20% |
| 31 | Europe | 0.27% | 0.29% | 0.27% | 2.36% | 2.50% | 2.39% | 11.38% | 12.02% | 11.61% |
| 32 | EU | 0.64% | 0.61% | 0.56% | 3.32% | 3.47% | 3.26% | 12.74% | 13.34% | 12.87% |
| 33 | WORLD | 0.12% | 0.13% | 0.13% | 0.99% | 1.05% | 1.02% | 4.71% | 5.16% | 4.94% |
| 34 | **Maximum** | 0.49% | 0.47% | 0.48% | 3.85% | 3.66% | 3.77% | 16.95% | 16.39% | 16.62% |
| 35 | **Average** | 0.30% | 0.29% | 0.29% | 2.33% | 2.27% | 2.28% | 11.12% | 10.89% | 10.97% |
| 36 | **Median** | 0.31% | 0.30% | 0.30% | 2.44% | 2.35% | 2.38% | 11.44% | 11.28% | 11.41% |
| 37 | **Minimum** | 0.10% | 0.05% | 0.06% | 0.82% | 0.67% | 0.78% | 4.17% | 3.42% | 3.87% |
| 38 | Row 20–row 33 | 0.12% | 0.16% | 0.15% | 0.92% | 1.24% | 1.18% | 4.33% | 5.62% | 5.52% |
| 39 | Row 20–row 35 | −0.06% | 0.00% | −0.01% | −0.42% | 0.02% | −0.08% | −2.08% | −0.11% | −0.52% |
| 40 | Row 20–row 36 | −0.12% | −0.11% | −0.12% | −0.94% | −0.89% | −0.95% | −3.45% | −3.51% | −3.78% |

| № | Percentile | p0p50 (Bottom 50%) | | | p50p90 (Middle 40%) | | | | | |
|---|---|---|---|---|---|---|---|---|---|---|
| | Country | 2009 | 2019 | 10Y Av. | 2009 | 2019 | 10Y Av. | | | |
| | 1 | 11 | 12 | 13 | 14 | 15 | 16 | | | |
| 1 | England | 17.77% | 20.36% | 19.43% | 43.74% | 43.94% | 44.01% | | | |
| 2 | Brazil | 10.36% | 9.81% | 10.75% | 30.80% | 30.89% | 32.07% | | | |
| 3 | Bulgaria | 19.90% | 16.50% | 18.27% | 45.13% | 40.02% | 43.09% | | | |
| 4 | Germany | 19.16% | 18.70% | 18.71% | 43.69% | 43.78% | 43.88% | | | |
| 5 | Denmark | 24.12% | 22.20% | 22.49% | 46.89% | 45.68% | 45.42% | | | |
| 6 | Israel | 13.42% | 14.75% | 14.17% | 35.96% | 37.84% | 37.10% | | | |
| 7 | India | 14.77% | 13.13% | 13.50% | 33.40% | 29.74% | 30.57% | | | |
| 8 | Spain | 20.18% | 20.73% | 20.71% | 44.55% | 44.53% | 44.88% | | | |
| 9 | Italy | 21.88% | 20.72% | 21.20% | 47.88% | 46.87% | 47.55% | | | |
| 10 | Canada | 16.74% | 15.58% | 16.25% | 43.91% | 43.72% | 43.27% | | | |
| 11 | Qatar | 12.44% | 12.75% | 12.69% | 34.97% | 33.91% | 34.11% | | | |
| 12 | China | 14.14% | 14.36% | 14.26% | 43.24% | 43.98% | 43.70% | | | |
| 13 | Luxembourg | 21.58% | 20.44% | 20.47% | 45.09% | 46.68% | 45.57% | | | |
| 14 | Mexico | 7.27% | 8.49% | 7.71% | 33.10% | 32.96% | 32.75% | | | |
| 15 | Netherlands | 23.56% | 22.14% | 22.68% | 47.84% | 47.96% | 48.06% | | | |
| 16 | New Zealand | 20.46% | 19.57% | 19.72% | 48.03% | 45.86% | 46.88% | | | |
| 17 | Norway | 25.04% | 23.93% | 24.16% | 44.09% | 44.45% | 44.11% | | | |
| 18 | Poland | 19.92% | 20.09% | 20.01% | 43.72% | 42.78% | 42.97% | | | |
| 19 | Romania | 15.37% | 15.14% | 16.13% | 39.75% | 43.40% | 42.78% | | | |

Table 5. *Cont.*

| № | Percentile | p0p50 (Bottom 50%) | | | p50p90 (Middle 40%) | | | | | |
|---|---|---|---|---|---|---|---|---|---|---|
| | Country | 2009 | 2019 | 10Y Av. | 2009 | 2019 | 10Y Av. | | | |
| | 1 | 11 | 12 | 13 | 14 | 15 | 16 | | | |
| 20 | **Russia** | **14.48%** | **16.98%** | **16.50%** | **35.91%** | **36.59%** | **37.03%** | | | |
| 21 | USA | 13.88% | 13.31% | 13.18% | 43.53% | 41.24% | 41.84% | | | |
| 22 | Turkmenistan | 13.05% | 13.05% | 13.11% | 38.03% | 38.03% | 38.12% | | | |
| 23 | Turkey | 14.61% | 15.48% | 15.31% | 33.93% | 33.82% | 33.84% | | | |
| 24 | Finland | 21.74% | 21.68% | 21.64% | 46.22% | 44.90% | 45.93% | | | |
| 25 | France | 21.75% | 21.63% | 21.68% | 46.11% | 46.14% | 46.12% | | | |
| 26 | Czech Republic | 25.77% | 25.39% | 25.42% | 45.07% | 45.53% | 45.03% | | | |
| 27 | Sweden | 24.10% | 24.49% | 24.15% | 46.20% | 46.23% | 46.07% | | | |
| 28 | South Africa | 8.99% | 5.80% | 6.44% | 31.71% | 28.79% | 29.34% | | | |
| 29 | Japan | 17.91% | 17.67% | 17.70% | 39.35% | 39.17% | 39.19% | | | |
| 30 | Asia | 10.03% | 10.23% | 9.99% | 37.36% | 39.14% | 38.48% | | | |
| 31 | Europe | 17.94% | 18.73% | 18.22% | 46.01% | 45.41% | 45.75% | | | |
| 32 | EU | 19.33% | 20.08% | 19.53% | 45.52% | 44.77% | 45.28% | | | |
| 33 | WORLD | 7.86% | 8.63% | 8.28% | 37.58% | 39.29% | 38.59% | | | |
| 34 | **Maximum** | **25.77%** | **25.39%** | **25.42%** | **48.03%** | **47.96%** | **48.06%** | | | |
| 35 | **Average** | **17.74%** | **17.41%** | **17.53%** | **41.44%** | **41.01%** | **41.22%** | | | |
| 36 | **Median** | **17.91%** | **17.67%** | **18.27%** | **43.72%** | **43.72%** | **43.27%** | | | |
| 37 | **Minimum** | **7.27%** | **5.80%** | **6.44%** | **30.80%** | **28.79%** | **29.34%** | | | |
| 38 | Row 20–row 33 | 6.62% | 8.35% | 8.22% | −1.67% | −2.70% | −1.55% | | | |
| 39 | Row 20–row 35 | −3.26% | −0.43% | −1.04% | −5.53% | −4.42% | −4.18% | | | |
| 40 | Row 20–row 36 | −4.03% | −4.36% | −5.09% | −0.19% | −2.48% | −1.43% | | | |

| № | Percentile | p80p100 (Top 20%) | | | p90p100 (Top 10%) | | | p99p100 (Top 1%) | | |
|---|---|---|---|---|---|---|---|---|---|---|
| | Country | 2009 | 2019 | 10Y Av. | 2009 | 2019 | 10Y Av. | 2009 | 2019 | 10Y Av. |
| | 1 | 17 | 18 | 19 | 20 | 21 | 22 | 23 | 24 | 25 |
| 1 | England | 53.85% | 50.68% | 51.65% | 38.49% | 35.71% | 36.56% | 14.64% | 12.93% | 13.38% |
| 2 | Brazil | 71.22% | 71.71% | 69.93% | 58.84% | 59.29% | 57.18% | 30.61% | 30.98% | 27.77% |
| 3 | Bulgaria | 50.20% | 57.66% | 53.44% | 34.97% | 43.49% | 38.64% | 11.45% | 18.24% | 14.48% |
| 4 | Germany | 52.07% | 52.50% | 52.44% | 37.15% | 37.51% | 37.40% | 12.93% | 13.03% | 13.01% |
| 5 | Denmark | 43.57% | 46.87% | 46.63% | 29.00% | 32.12% | 32.09% | 9.49% | 11.39% | 11.73% |
| 6 | Israel | 64.61% | 61.96% | 63.06% | 50.62% | 47.41% | 48.73% | 17.85% | 14.72% | 16.07% |
| 7 | India | 63.90% | 67.88% | 66.99% | 51.83% | 57.13% | 55.94% | 21.27% | 21.73% | 21.68% |
| 8 | Spain | 50.28% | 49.37% | 49.20% | 35.27% | 34.74% | 34.41% | 12.11% | 12.38% | 11.85% |
| 9 | Italy | 46.03% | 48.01% | 46.99% | 30.24% | 32.41% | 31.25% | 7.52% | 8.84% | 8.15% |
| 10 | Canada | 55.46% | 56.82% | 56.36% | 39.35% | 40.70% | 40.48% | 13.27% | 14.79% | 14.41% |
| 11 | Qatar | 66.86% | 67.09% | 67.05% | 52.59% | 53.33% | 53.20% | 18.70% | 19.46% | 19.32% |
| 12 | China | 59.02% | 58.36% | 58.64% | 42.63% | 41.66% | 42.04% | 15.52% | 14.00% | 14.26% |
| 13 | Luxembourg | 48.54% | 48.85% | 49.47% | 33.33% | 32.88% | 33.96% | 10.31% | 9.18% | 10.89% |
| 14 | Mexico | 74.68% | 73.01% | 74.26% | 59.63% | 58.55% | 59.54% | 27.11% | 28.71% | 28.01% |
| 15 | Netherlands | 44.09% | 45.76% | 45.03% | 28.60% | 29.90% | 29.26% | 6.55% | 7.06% | 6.79% |
| 16 | New Zealand | 45.58% | 48.12% | 47.51% | 31.51% | 34.57% | 33.40% | 9.94% | 11.87% | 10.85% |
| 17 | Norway | 44.41% | 45.73% | 45.61% | 30.87% | 31.62% | 31.73% | 10.90% | 10.68% | 11.08% |
| 18 | Poland | 51.21% | 51.41% | 51.46% | 36.36% | 37.14% | 37.02% | 13.55% | 14.58% | 14.18% |
| 19 | Romania | 54.54% | 57.20% | 55.98% | 44.88% | 41.46% | 41.08% | 18.37% | 14.45% | 15.05% |
| 20 | **Russia** | **62.43%** | **58.76%** | **59.19%** | **49.61%** | **46.43%** | **46.47%** | **21.32%** | **21.45%** | **20.63%** |
| 21 | USA | 58.32% | 60.63% | 60.35% | 42.59% | 45.46% | 44.98% | 16.72% | 18.76% | 18.49% |
| 22 | Turkmenistan | 63.52% | 63.52% | 63.38% | 48.92% | 48.92% | 48.77% | 18.99% | 18.99% | 18.87% |
| 23 | Turkey | 64.47% | 63.54% | 63.74% | 51.46% | 50.71% | 50.86% | 18.96% | 18.43% | 18.60% |
| 24 | Finland | 47.93% | 48.87% | 48.29% | 32.04% | 33.42% | 32.43% | 8.78% | 10.15% | 9.28% |
| 25 | France | 47.34% | 47.32% | 47.33% | 32.14% | 32.23% | 32.20% | 9.95% | 10.03% | 10.11% |
| 26 | Czech Republic | 42.67% | 42.61% | 43.05% | 29.16% | 29.08% | 29.55% | 10.87% | 10.37% | 10.58% |
| 27 | Sweden | 44.24% | 43.58% | 44.11% | 29.69% | 29.28% | 29.78% | 9.17% | 9.09% | 9.51% |
| 28 | South Africa | 74.26% | 79.55% | 78.54% | 59.29% | 65.41% | 64.22% | 18.58% | 19.31% | 19.14% |
| 29 | Japan | 56.55% | 56.97% | 56.92% | 42.74% | 43.16% | 43.11% | 11.35% | 11.62% | 11.58% |
| 30 | Asia | 68.05% | 66.80% | 67.49% | 52.61% | 50.63% | 51.53% | 20.86% | 18.49% | 19.61% |
| 31 | Europe | 51.97% | 51.41% | 51.79% | 36.05% | 35.86% | 36.03% | 11.55% | 11.78% | 11.71% |
| 32 | EU | 50.80% | 50.39% | 50.70% | 35.15% | 35.15% | 35.19% | 10.86% | 11.18% | 10.99% |
| 33 | WORLD | 71.23% | 68.94% | 69.84% | 54.56% | 52.08% | 53.14% | 19.75% | 19.34% | 19.60% |
| 34 | **Maximum** | **74.68%** | **79.55%** | **78.54%** | **59.63%** | **65.41%** | **64.22%** | **30.61%** | **30.98%** | **28.01%** |
| 35 | **Average** | **55.24%** | **56.01%** | **55.74%** | **40.82%** | **41.58%** | **41.25%** | **14.72%** | **15.08%** | **14.82%** |
| 36 | **Median** | **53.85%** | **56.82%** | **53.44%** | **38.49%** | **40.70%** | **38.64%** | **13.27%** | **14.00%** | **14.18%** |
| 37 | **Minimum** | **42.67%** | **42.61%** | **43.05%** | **28.60%** | **29.08%** | **29.26%** | **6.55%** | **7.06%** | **6.79%** |
| 38 | Row 20–row 33 | −8.80% | −10.18% | −10.65% | −4.95% | −5.65% | −6.67% | 1.57% | 2.11% | 1.02% |
| 39 | Row 20–row 35 | 7.19% | 2.75% | 3.45% | 8.79% | 4.85% | 5.22% | 6.60% | 6.37% | 5.81% |
| 40 | Row 20–row 36 | 4.47% | 3.81% | 6.91% | 4.10% | 4.76% | 6.34% | 3.45% | 4.76% | 4.30% |

Source: World Inequality Database (2020a).

The comparison of Russia with the world averages (line 36 in Table 5) provides us with the following information:

1. Problem of poverty in Russia is less than in many other counties. A larger share of income is distributed the lower 50% of Russian households compared with other countries (columns 2–13 in Table 5).
2. The middle class in Russia (columns 14–16 in Table 5) has a lot in common with countries from the group of BRICSs, but at the same time it is much smaller than in developed countries with a high national income per capita.
3. On the other hand, the top 20%, top 10%, and top 1% income groups in Russia are dominating in distribution. The structure of the right tail of income distribution in Russia is very similar to such developing countries as Brazil, Mexico, South Africa, India and, to some extent, the USA. All of these countries have a convex group of rich households, exceeding, by their size, world averages. The top 20% and even top 10% of Russian households earn, on average, a slightly lower share of the national income than the world average (row 36 and columns 17–22 in Table 5). However, the top 1% of Russia's richest households earn more than 20% of the national income, which is 1.5%−2% more than the global average. That indicates extreme inequality in the top of income distribution in Russia. The picture of top income inequality is worse only in three countries from our sample—Brazil, Mexico, and India.

The comparison of Russia with the average and median values of our 27-country sample (lines 37 and 38 in Table 5) provides us with a slightly different picture from what we saw before, in line 36 of Table 5.

4. Russia is a poorer and more unfair country in terms of income distribution because the bottom 50% of households earn less than the average and median of our sample. This means that when we weed out countries with very high levels of income inequality from the averages in line 31 of Table 5, Russia looks much worse against the backdrop of the developed part of the world.
5. Income distribution is still highly skewed in favor of the top 20% of households. So, the overall inequality picture is even worse than it is in line 31 of Table 5.

The ratio analysis also shows the problem of income concentration at the top of distribution (Table 6). The gap between the upper decile group and the various components of the lower 50% in Russia is significant.

**Table 6.** Ratio analysis of pre-tax income distribution by household groups.

| № | Percentile | p50p90/p0p50 (Middle 40%/Bottom 50%) | | | p80p100/p0p20 (Quintile Ratio) | | | p80p100/p0p50 (Top 20%/Bottom 50%) | | |
|---|---|---|---|---|---|---|---|---|---|---|
| | Country | 2009 | 2019 | 10Y Av. | 2009 | 2019 | 10Y Av. | 2009 | 2019 | 10Y Av. |
| | 1 | 2 | 3 | 4 | 5 | 6 | 7 | 8 | 9 | 10 |
| 1 | England | 2.46 | 2.16 | 2.27 | 24.15 | 18.50 | 20.51 | 3.03 | 2.49 | 2.66 |
| 2 | Brazil | 2.97 | 3.15 | 2.99 | 67.19 | 103.93 | 73.79 | 6.87 | 7.31 | 6.53 |
| 3 | Bulgaria | 2.27 | 2.43 | 2.36 | 19.53 | 25.97 | 22.41 | 2.52 | 3.49 | 2.92 |
| 4 | Germany | 2.28 | 2.34 | 2.35 | 21.34 | 22.34 | 22.42 | 2.72 | 2.81 | 2.80 |
| 5 | Denmark | 1.94 | 2.06 | 2.02 | 13.12 | 15.17 | 14.90 | 1.81 | 2.11 | 2.08 |
| 6 | Israel | 2.68 | 2.57 | 2.62 | 37.56 | 32.61 | 34.68 | 4.81 | 4.20 | 4.46 |
| 7 | India | 2.26 | 2.27 | 2.26 | 31.17 | 37.30 | 35.85 | 4.33 | 5.17 | 4.97 |
| 8 | Spain | 2.21 | 2.15 | 2.17 | 18.49 | 18.08 | 18.28 | 2.49 | 2.38 | 2.38 |
| 9 | Italy | 2.19 | 2.26 | 2.24 | 16.09 | 18.39 | 17.50 | 2.10 | 2.32 | 2.22 |
| 10 | Canada | 2.62 | 2.81 | 2.67 | 23.11 | 25.83 | 24.36 | 3.31 | 3.65 | 3.47 |
| 11 | Qatar | 2.81 | 2.66 | 2.69 | 41.02 | 37.69 | 38.39 | 5.37 | 5.26 | 5.28 |
| 12 | China | 3.06 | 3.06 | 3.07 | 37.35 | 36.48 | 37.11 | 4.17 | 4.06 | 4.11 |
| 13 | Luxembourg | 2.09 | 2.28 | 2.23 | 16.23 | 18.50 | 18.19 | 2.25 | 2.39 | 2.42 |
| 14 | Mexico | 4.55 | 3.88 | 4.27 | 91.07 | 70.88 | 82.47 | 10.27 | 8.60 | 9.68 |
| 15 | Netherlands | 2.03 | 2.17 | 2.12 | 14.18 | 16.11 | 15.23 | 1.87 | 2.07 | 1.99 |
| 16 | New Zealand | 2.35 | 2.34 | 2.38 | 16.76 | 18.51 | 18.09 | 2.23 | 2.46 | 2.41 |
| 17 | Norway | 1.76 | 1.86 | 1.83 | 12.65 | 13.61 | 13.49 | 1.77 | 1.91 | 1.89 |
| 18 | Poland | 2.19 | 2.13 | 2.15 | 19.70 | 19.85 | 19.93 | 2.57 | 2.56 | 2.57 |
| 19 | Romania | 2.59 | 2.87 | 2.65 | 29.97 | 32.32 | 33.54 | 3.55 | 3.78 | 3.47 |

**Table 6.** *Cont.*

| № | Percentile | p50p90/p0p50 (Middle 40%/Bottom 50%) | | | p80p100/p0p20 (Quintile Ratio) | | | p80p100/p0p50 (Top 20%/Bottom 50%) | | |
|---|---|---|---|---|---|---|---|---|---|---|
| | Country | 2009 | 2019 | 10Y Av. | 2009 | 2019 | 10Y Av. | 2009 | 2019 | 10Y Av. |
| | 1 | 2 | 3 | 4 | 5 | 6 | 7 | 8 | 9 | 10 |
| 20 | **Russia** | 2.48 | 2.15 | 2.25 | 32.69 | 25.66 | 26.97 | 4.31 | 3.46 | 3.60 |
| 21 | USA | 3.14 | 3.10 | 3.18 | 38.88 | 41.53 | 42.08 | 4.20 | 4.56 | 4.58 |
| 22 | Turkmenistan | 2.91 | 2.91 | 2.91 | 39.70 | 39.70 | 40.24 | 4.87 | 4.87 | 4.84 |
| 23 | Turkey | 2.32 | 2.18 | 2.21 | 32.40 | 29.42 | 29.95 | 4.41 | 4.10 | 4.17 |
| 24 | Finland | 2.13 | 2.07 | 2.12 | 16.03 | 16.24 | 16.25 | 2.20 | 2.25 | 2.23 |
| 25 | France | 2.12 | 2.13 | 2.13 | 17.34 | 17.66 | 17.54 | 2.18 | 2.19 | 2.18 |
| 26 | Czech Republic | 1.75 | 1.79 | 1.77 | 11.08 | 11.64 | 11.44 | 1.66 | 1.68 | 1.69 |
| 27 | Sweden | 1.92 | 1.89 | 1.91 | 13.65 | 13.09 | 13.47 | 1.84 | 1.78 | 1.83 |
| 28 | South Africa | 3.53 | 4.96 | 4.64 | 60.37 | 118.73 | 105.41 | 8.26 | 13.72 | 12.52 |
| 29 | Japan | 2.20 | 2.22 | 2.21 | 22.80 | 23.35 | 23.27 | 3.16 | 3.22 | 3.22 |
| 30 | Asia | 3.72 | 3.83 | 3.85 | 48.96 | 47.38 | 48.89 | 6.78 | 6.53 | 6.76 |
| 31 | Europe | 2.56 | 2.42 | 2.51 | 22.02 | 20.56 | 21.65 | 2.90 | 2.74 | 2.84 |
| 32 | EU | 2.35 | 2.23 | 2.32 | 15.30 | 14.52 | 15.58 | 2.63 | 2.51 | 2.60 |
| 33 | WORLD | 4.78 | 4.55 | 4.67 | 71.95 | 65.66 | 68.41 | 9.06 | 7.99 | 8.45 |
| 34 | **Maximum** | 4.55 | 4.96 | 4.64 | 91.07 | 118.73 | 105.41 | 10.27 | 13.72 | 12.52 |
| 35 | **Average** | 2.48 | 2.51 | 2.51 | 28.81 | 31.69 | 30.61 | 3.63 | 3.82 | 3.77 |
| 36 | **Median** | 2.28 | 2.27 | 2.26 | 22.80 | 23.35 | 22.42 | 3.03 | 3.22 | 2.92 |
| 37 | **Minimum** | 1.75 | 1.79 | 1.77 | 11.08 | 11.64 | 11.44 | 1.66 | 1.68 | 1.69 |

| № | Percentile | p90p100/p0p10 (Decile Ratio) | | | p90p100/p0p40 (Palm Ratio) | | |
|---|---|---|---|---|---|---|---|
| | Country | 2009 | 2019 | 10Y Av. | 2009 | 2019 | 10Y Av. |
| | 1 | 11 | 12 | 13 | 14 | 15 | 16 |
| 1 | England | 132.72 | 102.03 | 113.49 | 3.51 | 2.76 | 3.01 |
| 2 | Brazil | 309.68 | 423.50 | 329.45 | 9.48 | 10.74 | 9.08 |
| 3 | Bulgaria | 105.97 | 155.32 | 127.25 | 2.82 | 4.20 | 3.39 |
| 4 | Germany | 119.84 | 125.03 | 125.23 | 3.09 | 3.23 | 3.22 |
| 5 | Denmark | 69.05 | 82.36 | 80.71 | 1.90 | 2.27 | 2.24 |
| 6 | Israel | 230.09 | 197.54 | 210.89 | 6.12 | 5.20 | 5.59 |
| 7 | India | 199.35 | 248.39 | 236.39 | 5.47 | 6.79 | 6.48 |
| 8 | Spain | 100.77 | 99.26 | 99.91 | 2.77 | 2.66 | 2.64 |
| 9 | Italy | 81.73 | 98.21 | 91.16 | 2.20 | 2.52 | 2.37 |
| 10 | Canada | 126.94 | 145.36 | 136.81 | 3.72 | 4.17 | 3.96 |
| 11 | Qatar | 250.43 | 231.87 | 236.22 | 6.76 | 6.58 | 6.61 |
| 12 | China | 213.15 | 208.30 | 211.23 | 5.10 | 4.89 | 4.99 |
| 13 | Luxembourg | 87.71 | 96.71 | 98.03 | 2.40 | 2.56 | 2.63 |
| 14 | Mexico | 331.28 | 254.57 | 292.01 | 14.30 | 11.71 | 13.34 |
| 15 | Netherlands | 71.50 | 83.06 | 77.69 | 1.92 | 2.16 | 2.06 |
| 16 | New Zealand | 90.03 | 104.76 | 99.98 | 2.52 | 2.89 | 2.77 |
| 17 | Norway | 68.60 | 73.53 | 73.54 | 1.91 | 2.05 | 2.03 |
| 18 | Poland | 110.18 | 112.55 | 112.81 | 2.91 | 2.96 | 2.96 |
| 19 | Romania | 195.13 | 180.26 | 228.24 | 4.82 | 4.54 | 4.29 |
| 20 | **Russia** | 206.71 | 160.10 | 166.78 | 5.49 | 4.31 | 4.46 |
| 21 | USA | 224.16 | 239.26 | 243.96 | 5.33 | 5.85 | 5.90 |
| 22 | Turkmenistan | 244.60 | 244.60 | 243.87 | 6.11 | 6.11 | 6.07 |
| 23 | Turkey | 205.84 | 181.11 | 185.49 | 5.53 | 5.08 | 5.17 |
| 24 | Finland | 84.32 | 87.95 | 86.19 | 2.31 | 2.40 | 2.34 |
| 25 | France | 91.83 | 94.79 | 93.99 | 2.36 | 2.39 | 2.38 |
| 26 | Czech Republic | 59.51 | 61.87 | 61.60 | 1.72 | 1.77 | 1.78 |
| 27 | Sweden | 72.41 | 69.71 | 71.46 | 1.93 | 1.87 | 1.93 |
| 28 | South Africa | 592.90 | 1308.20 | 1098.73 | 10.44 | 19.13 | 17.20 |
| 29 | Japan | 133.56 | 139.23 | 138.68 | 3.74 | 3.83 | 3.82 |
| 30 | Asia | 350.73 | 337.53 | 345.82 | 8.39 | 7.97 | 8.31 |
| 31 | Europe | 133.52 | 123.66 | 133.60 | 3.17 | 2.98 | 3.10 |
| 32 | EU | 54.92 | 57.62 | 63.01 | 2.76 | 2.63 | 2.74 |
| 33 | WORLD | 454.67 | 400.62 | 421.37 | 11.58 | 10.09 | 10.78 |
| 34 | **Maximum** | 592.90 | 1308.20 | 1098.73 | 14.30 | 19.13 | 17.20 |
| 35 | **Average** | 165.86 | 193.43 | 185.24 | 4.44 | 4.75 | 4.65 |
| 36 | **Median** | 126.94 | 139.23 | 127.25 | 3.51 | 3.83 | 3.39 |
| 37 | **Minimum** | 59.51 | 61.87 | 61.60 | 1.72 | 1.77 | 1.78 |

<div align="center">

**Table 6.** *Cont.*

</div>

| № | Percentile | p90p100/p0p50 (Top 10%/Bottom 50%) | | | p99p100/p90p100 (Top 1%/Top 10%) | | | |
|---|---|---|---|---|---|---|---|---|
| | Country | 2009 | 2019 | 10Y Av. | 2009 | 2019 | 10Y Av. | |
| | 1 | 17 | 18 | 19 | 20 | 21 | 22 | |
| 1 | England | 2.17 | 1.75 | 1.89 | 0.38 | 0.36 | 0.37 | |
| 2 | Brazil | 5.68 | 6.04 | 5.34 | 0.52 | 0.52 | 0.48 | |
| 3 | Bulgaria | 1.76 | 2.64 | 2.11 | 0.33 | 0.42 | 0.37 | |
| 4 | Germany | 1.94 | 2.01 | 2.00 | 0.35 | 0.35 | 0.35 | |
| 5 | Denmark | 1.20 | 1.45 | 1.43 | 0.33 | 0.35 | 0.37 | |
| 6 | Israel | 3.77 | 3.21 | 3.45 | 0.35 | 0.31 | 0.33 | |
| 7 | India | 3.51 | 4.35 | 4.16 | 0.41 | 0.38 | 0.39 | |
| 8 | Spain | 1.75 | 1.68 | 1.66 | 0.34 | 0.36 | 0.34 | |
| 9 | Italy | 1.38 | 1.56 | 1.48 | 0.25 | 0.27 | 0.26 | |
| 10 | Canada | 2.35 | 2.61 | 2.49 | 0.34 | 0.36 | 0.36 | |
| 11 | Qatar | 4.23 | 4.18 | 4.19 | 0.36 | 0.36 | 0.36 | |
| 12 | China | 3.01 | 2.90 | 2.95 | 0.36 | 0.34 | 0.34 | |
| 13 | Luxembourg | 1.54 | 1.61 | 1.66 | 0.31 | 0.28 | 0.32 | |
| 14 | Mexico | 8.20 | 6.90 | 7.76 | 0.45 | 0.49 | 0.47 | |
| 15 | Netherlands | 1.21 | 1.35 | 1.29 | 0.23 | 0.24 | 0.23 | |
| 16 | New Zealand | 1.54 | 1.77 | 1.70 | 0.32 | 0.34 | 0.32 | |
| 17 | Norway | 1.23 | 1.32 | 1.31 | 0.35 | 0.34 | 0.35 | |
| 18 | Poland | 1.83 | 1.85 | 1.85 | 0.37 | 0.39 | 0.38 | |
| 19 | Romania | 2.92 | 2.74 | 2.55 | 0.41 | 0.35 | 0.37 | |
| **20** | **Russia** | 3.43 | 2.73 | 2.83 | 0.43 | 0.46 | 0.44 | |
| 21 | USA | 3.07 | 3.42 | 3.42 | 0.39 | 0.41 | 0.41 | |
| 22 | Turkmenistan | 3.75 | 3.75 | 3.72 | 0.39 | 0.39 | 0.39 | |
| 23 | Turkey | 3.52 | 3.28 | 3.32 | 0.37 | 0.36 | 0.37 | |
| 24 | Finland | 1.47 | 1.54 | 1.50 | 0.27 | 0.30 | 0.29 | |
| 25 | France | 1.48 | 1.49 | 1.49 | 0.31 | 0.31 | 0.31 | |
| 26 | Czech Republic | 1.13 | 1.15 | 1.16 | 0.37 | 0.36 | 0.36 | |
| 27 | Sweden | 1.23 | 1.20 | 1.23 | 0.31 | 0.31 | 0.32 | |
| 28 | South Africa | 6.60 | 11.28 | 10.25 | 0.31 | 0.30 | 0.30 | |
| 29 | Japan | 2.39 | 2.44 | 2.44 | 0.27 | 0.27 | 0.27 | |
| 30 | Asia | 5.25 | 4.95 | 5.16 | 0.40 | 0.37 | 0.38 | |
| 31 | Europe | 2.01 | 1.91 | 1.98 | 0.32 | 0.33 | 0.32 | |
| 32 | EU | 1.82 | 1.75 | 1.80 | 0.31 | 0.32 | 0.31 | |
| 33 | WORLD | 6.94 | 6.03 | 6.43 | 0.36 | 0.37 | 0.37 | |
| 34 | **Maximum** | 8.20 | 11.28 | 10.25 | 0.52 | 0.52 | 0.48 | |
| 35 | **Average** | 2.73 | 2.90 | 2.85 | 0.35 | 0.35 | 0.35 | |
| 36 | **Median** | 2.17 | 2.44 | 2.11 | 0.35 | 0.35 | 0.36 | |
| 37 | **Minimum** | 1.13 | 1.15 | 1.16 | 0.23 | 0.24 | 0.23 | |

<div align="center">

Source: World Inequality Database (2020a).

</div>

## 5. Discussion

It is obvious that the overall level of income inequality in Russia is not the highest in the world. There are many countries in which income differentiation that exceeds Russia's. The commonly known stereotype, i.e., Russia is an extremely inequal country is incorrect if we compare it with world averages. The dynamics of income inequality in the Rosstat database are very similar with those in the World inequality database.

Russia is significantly more inequal in market income in comparison with the developed countries of continental Europe. Russian households are poorer at the left tail of income distribution and in the center. The top 10% of households in Russia receive a much higher share of the country's national income than in most other countries in our sample.

In terms of poverty, Russia looks better than many comparable countries with similarly structured economies and exports. At the same time, Russia is noticeably lagging behind most developed countries in combating poverty despite the very impressive government activities of reducing poverty in Russia after 2005.

Russia has a problem of an excessive income inequality at the top of distribution. According to all indicators, the largest concentration of market income is concentrated in the hands of the top 1% of Russians. It is noteworthy that we cannot find this problem from

the Rosstat data. It is not visible in Rosstat's official point of view on income inequality in Russia, as Rosstat simply does not show the top of income distribution in detail.

The national income share of Russia's top 20% income group in the World inequality database in 2019 is more than 10% higher than in the Rosstat database (58.8% vs. 46.9%). It makes sense to return to the problem of the low quality of data. Earlier in the article, we mentioned the research of Kapeliushnikov (2020) wherein he wrote that extreme inequality at the top of the income distribution in Russia may be an exaggeration, arising from inaccurate data from the World inequality database. Of course, this hypothesis should be taken into account in the evaluation of the inequality in Russia. First of all, we would like to emphasize that, according to the World inequality database, the problem of extreme inequality at the top of the income distribution in Brazil, Mexico, and South Africa is even worse than in Russia. T. Piketty team points out the lower inequality transparency in developing countries (World Inequality Database 2020a) and Russia is in the list of these countries. In this regard, Rosstat should improve the transparency of socio-economic inequality in Russia, implement best practices, and improve its methodology to world standards. This will strengthen the official position concerning income inequality in Russia, according to which Russia has no significant problems in this sphere of social-economic development. Additionally, this progress can help to confirm or refute if the thesis of excessive differentiation of income in favor of the richest Russians is true or false (An et al. 2021; Mikhaylov 2021; Mutalimov et al. 2021).

To clarify the issue of income data inconsistencies in different sources, we propose to switch the focus from income onto wealth inequality in Russia. These indicators of socio-economic development of a county have much in common and are very interrelated with each other. Wealth inequality could be interpreted as a disbalance in stock accumulation which derives from differentiation in income distribution, as flows of value of income between economic agents (Kinsella et al. 2011; Caverzasi and Godin 2014; Detzer 2016; Caiani et al. 2016a, 2016b). We can assume that a country with high income inequality is becoming more inequal in wealth in long-term, because the flow of income goes to the top of distribution and increases their stock of wealth significantly over time. Comparing the level and dynamics of wealth in different countries, we can indirectly evaluate the scale of income inequality in a country.

According to Credit Suisse Group AG (2020), Russia has an outstanding problem of wealth inequality. Extreme levels level of wealth inequality in Russia prove that reforms to financial deregulation of the economy in the 2000s were incorrect decisions in terms of managing the problem of socio-economic inequality. Historical dynamics of income inequality according to Rosstat also shows that the processes of wealth and income distribution after the collapse of the USSR were unfair. Basing on this logic, we tend to think that real figures of income inequality in Russia are higher than the official Rosstat's estimates and are probably somewhere in the middle between Rosstat's and T. Pikkety's team view. At the same time, the data obtained from the independent source (Credit Suisse Group AG 2020) show the problem of wealth inequality at the top, so it is very likely that Russia has the same problem of income inequality as well.

Considering all the facts, we should say that current income inequality in Russia is not fatal for its sustainable development. The Russian government should adjust the redistribution of national income from the top 10% income group to the center (p40-p90), focusing on optimizing a progressive taxation system for the richest 1% of households. Success in this area will be an additional factor in ensuring the sustainable and harmonious economic growth of Russia.

## 6. Conclusions

The article analyzed the historical dynamics and modern level of income inequality in Russia. The analysis of Rosstat's database showed that the level of income inequality in Russia increased sharply after the collapse of the USSR and increased until 2008. After that, the level of income inequality began a steady decline. Rosstat data are not suitable

for cross-country comparisons of Russia due to significant differences in the accounting methodology and transparency of data in Russia. Rosstat does not show the structure of the top 10% of income distribution in Russia.

The results of opinion polls concerning economic inequality show that Russian households rate the level of socio-economic inequality as high. Nonetheless, the thesis that Russia has extremely high inequality in income distribution, in reality, is not correct according to any source of statistical data if we compare Russia with other countries.

The use of the World inequality database helped us to make cross-country comparisons of income inequality in Russia via building heatmaps. The research gave us information suggesting Russia has much in common in income distribution with such countries as Brazil, India, Mexico, South Africa, and to a lesser extent the United States. All of these countries face the problem of top income inequality.

There are many countries in the world economy wherein the differentiation of income in which significantly exceeds Russia. At the same time, if Russia wants to build a social state comparable with EU countries, it has to implement many more reforms in all areas of socio-economic development in the future.

To enter the trajectory of sustainable economic growth, the Russian Government should continue raising the marginal tax rates on personal income and wealth of the top 1%, bringing the progressive tax system of Russia closer to the standards of developed countries (Simula and Trannoy 2020). It is essential to do this gradually and avoid sudden and potentially destabilizing hikes of tax rates, because ill-conceived decisions can slow down the economic growth of the Russian economy. At the same time, it is also recommended to decrease the tax burden of the bottom 40% income group via increasing the volume of transfers from the federal budget, developing the system of personal tax deductions, or implementing regressive personal income tax rates.

The advantages of comparative analysis based on the construction of income inequality heatmaps are their simplicity, informativity, and detailing. The vast majority of econometric research methods do not provide such a depth of household income distribution understanding. However, it is obvious that the limitations of comparative analysis via building income inequality heatmaps method are its bulkiness and lengthiness compared with slender and concise econometric models. The inclusion of a simple set of statistical averages (maximum, minimum, average, and median) strengthens the effectiveness of a heatmap as a method of comparative valuation of income inequality. From our point of view, this method increases the potential and strengthens the analytical depth of econometric and econophysics models, which are typically used in the research of income inequality.

The objective of the study was to reveal the details of household income distribution in different countries and to demonstrate the degree of their differentiation by building several income inequality heatmaps. This objective is better achieved by using income inequality heatmaps, because the method gives us gives us the required details. At the same time, econometric estimation techniques could be used in further investigation as an addition to inequality heatmaps. A particular econometric model or an integral statistical indicator could be used as input data for building an income inequality heatmap in different countries. Additionally, further development of comparative analysis with the use of inequality heatmaps, which were used in this research, could be valuable in enhancing the realism of theoretical econophysics models by considering real initial incoming data with the distribution of inequality in different countries before conducting AB-SCF modeling procedures (Caiani et al. 2016a, 2016b; Godley and Lavoie 2007).

**Funding:** This research received no external funding.

**Institutional Review Board Statement:** Not applicable.

**Informed Consent Statement:** Not applicable.

**Conflicts of Interest:** The author declares no conflict of interest.

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
