# Peer review of "Does Income Inequality Create Excessive Threats to the Sustainable Development of Russia? Evidence from Intercountry Comparisons via Analysis of Inequality Heatmaps"

_economies, doi:10.3390/economies9040166_

Round 1

Reviewer 1 Report

Two elements need to be addressed. Firstly, the methodology should address the way some countries are excluded from the analysis, for example: Bulgaria and Romania. Secondly, the Conclusion section should be extended by adding a couple of words on the overall limits of the model and also what future enquiries could be done.

Reviewer 2 Report

Comments Summary: This paper examine that the income inequality in Russia in the context of sustainable development of Russia. The purpose of this research is to assess Russian specifics of income inequality and answer the question if income inequality in Russia is excessively high and needs extra government regulation in order to reach the trajectory of advanced sustainable development. Comments: 1. This is an interesting study. 2. The literature review is limited. The introduction focuses exclusively on the literature in Russia without having indicative references to what is happening in other countries. The literature review should be enriched. I recommend the authors extend the literature review. 3. The paper should further analyze the comparative method of assessment with building heatmaps of household income distributions Rosstat methodology. 4. The paper would be upgraded if the research question is investigated based on econometric estimation techniques.

Round 2

Reviewer 2 Report

Comments

The authors have made are corrections in the right direction.

Please adds the following papers that are more important:

  • Konstantakopoulou, I. Further Evidence on Import Demand Function and Income Inequality. Economies 2020, 8, 91. https://doi.org/10.3390/economies8040091.